# Analyzing Precision Medicine Utilization with Real-World Data: A Scoping Review

**DOI:** 10.3390/jpm12040557

**Published:** 2022-04-01

**Authors:** Michael P. Douglas, Anika Kumar

**Affiliations:** 1Center for Translational and Policy Research on Precision Medicine (TRANSPERS), Department of Clinical Pharmacy, University of California, San Francisco, San Francisco, CA 94143, USA; 2School of Medicine, University of California, San Francisco, San Francisco, CA 94143, USA; anika.kumar@ucsf.edu

**Keywords:** real-world data, precision medicine, registry, utilization, scoping review

## Abstract

Precision medicine (PM), specifically genetic-based testing, is currently used in over 140,000 individual tests to inform the clinical management of disease. Though several databases (e.g., the NIH Genetic Testing Registry) demonstrate the availability of these sequencing-based tests, we do not currently understand the extent to which these tests are used. There exists a need to synthesize the body of real-world data (RWD) describing the use of sequencing-based tests to inform their appropriate use. To accomplish this, we performed a scoping review to examine what RWD sources have been used in studies of PM utilization between January 2015 and August 2021 to characterize the use of genome sequencing (GS), exome sequencing (ES), tumor sequencing (TS), next-generation sequencing-based panels (NGS), gene expression profiling (GEP), and pharmacogenomics (PGx) panels. We abstracted variables describing the use of these types of tests and performed a descriptive statistical analysis. We identified 440 articles in our search and included 72 articles in our study. Publications based on registry databases were the most common, followed by studies based on private insurer administrative claims. Slightly more than one-third (38%) used integrated datasets. Two thirds (67%) of the studies focused on the use of tests for oncological clinical applications. We summarize the RWD sources used in peer-reviewed literature on the use of PM. Our findings will help improve future study design by encouraging the use of centralized databases and registries to track the implementation and use of PM.

## 1. Introduction

Since the sequencing of the first human genome, there has been an explosion in the development of precision medicine (PM) tests. This, in conjunction with the rapid reduction in the cost of genomic sequencing over the 15 years from approximately USD 14 million in 2006 to USD 1000 in 2021 [1], as well as the rise in bioinformatic and digital data-driven approaches to health care delivery [2], puts our health care system in a position to utilize PM technologies with significantly greater ubiquity and effectiveness. Therefore, there is a need to understand how PM is being used.

The implementation of PM is increasing, and thus it is important to understand how it is used in routine clinical care [3]. Achieving this requires the use of real-world data (RWD): data related to patient health states and/or the delivery of care routinely collected from a variety of sources, e.g., electronic health records (EHR), claims data, registries, etc. However, it is well known that the use of RWD presents numerous challenges (e.g., data validity and quality relating to unstructured data, haphazard and unstandardized collection, methods of obtaining RWD, and issues such as data access, privacy, and the ability to combine data), and these challenges may be magnified in the context of PM [4,5,6].

We define PM as the use of genetic testing to target interventions, including the use of genomic tests for diagnosis, screening in asymptomatic patients, risk assessment, determining the prognosis of a diagnosed disease, and predicting treatment responses or adverse events [7]. For example, the interrogation of the single genes *BRCA1* and *BRCA2* for the risk assessment of hereditary breast and ovarian cancer (HBOC) has been available since 1996 [8], and the use of whole-exome and whole-genome sequencing in the diagnosis of suspected genetic diseases has developed much more recently. While these tests are being used, the extent of their utilization individually or as a whole, and their impact on clinical management worldwide, is unknown [3,9]. One reason for this is the lack of a central repository of these data.

At present, there is no centrally integrated source of utilizable data (e.g., tests performed with the intent of using the results in clinical management) in the real world. For example, some data can be found in the gray literature (e.g., white papers, health system reports, regulatory filings, company websites, news reports, and national/international consortia websites), and some data can be obtained from administrative/clinical resources (e.g., EHR, claims data, fee schedules, industry databases, and registries) [3]. However, very few of the data records are linked, highlighting a need to synthesize real-world data (RWD) describing the use of PM sequencing-based tests in order to obtain an overall utilization picture and to understand their impact on clinical management.

Our objective was to identify where PM use data have been described in peer-reviewed publications and how they may be linked in the real world to track utilization. To accomplish this, we performed a scoping review to examine what real-world data sources have been used in studies of PM utilization. We add to the literature by providing a systematic assessment of the real-world databases and registries that have been used in existing peer-reviewed studies. Our findings can be used to inform future study design and encourage the development and use of centralized databases and registries to track the real-world use of PM-based tests. These centralized databases or registries would inform future studies on appropriate test utilization.

## 2. Materials and Methods

We sought to identify peer-reviewed publications that used RWD sources to describe the use of PM using a scoping review. We searched PubMed for articles written in English and published between January 2015 and August 2021 (Appendix A). We chose to limit our search to 5 years plus the current year, as this would identify the majority of articles on the topic (only nine articles would have been identified in the search prior to 2015, none of which met the inclusion criteria).

### 2.1. Inclusion/Exclusion

We included original research studies that examined PM, as defined above (e.g., whole-exome/genome sequencing (WES/WGS), tumor sequencing, next-generation sequencing (NGS) panels, gene expression profiling, pharmacogenetics, and hereditary cancer panels, including *BCRA1/2*), and its utilization (number and/or type of genetic tests performed) using real-world data (RWD; data related to patient health states and/or the delivery of care routinely collected from a variety of sources, e.g., electronic health records, claims data, registries, laboratories, or integrated datasets from these sources). We excluded non-genetic tests and single-gene test studies (while relevant, we wanted to focus on multi-gene tests, as those are more difficult to identify in databases such as claims databases due to coding issues), studies that did not fit our RWD definition (as defined above), and studies that were focused on economic evaluation versus test utilization.

### 2.2. Data Abstraction

We developed a data abstraction spreadsheet (Appendix A), and data were extracted independently by the two co-authors (A.K., M.P.D.). Discrepancies in coding, such as details regarding database names or test names or if datasets were integrated, were minimal and were resolved by discussion. We categorized studies using the following six categories (items in paratheses correspond to data abstracted):Demographic (PubMed ID, first author, year);Research question (research question);Name of RWD sources and whether or not the study used an integrated or merged dataset from two or more sources (clinical data source, admin claims (commercial) source, admin claims (public) source, registry source);Laboratory name that performed the PM testing, if available (lab name, other);Condition or disease, including if it was cancer or non-cancer condition (condition);Test name and/or genes, including if the test was BRCA, gene expression profiling/Oncotype Dx, multigene pane, tumor sequencing, WES/WGS, or other (test name and/or genes).

### 2.3. Analysis

Descriptive statistics were used to analyze data across variables.

## 3. Results

We identified 440 articles in our search and excluded 368, leaving a remaining 72 articles to be included in our study (Figure 1, Appendix A. Preferred Reporting Items for Systematic reviews and Meta-Analyses extension for Scoping Reviews (PRISMA-ScR) Checklist). We analyzed abstracted variables to determine the type of databases used, the laboratories used, if the condition/disease was cancer/non-cancer, and the type of test.

The two most common RWD types were administrative claims data (58%) and registry data (55%) (Table 1) [10]. Commonly used data sources were the SEER-Medicare (Surveillance, Epidemiology, and End Results Program-Medicare) linked database, the National Cancer Database, and Truven Health Analytics claims databases (now part of IBM Watson Health, Cambridge, MA, USA). Of importance is that over one-third (38%) of studies used an integrated data source (e.g., SEER-Medicare) or combined data from more than one source for their analyses. The use of such datasets was often in response to the challenges of using RWD from a single source. For example, several studies linked data from laboratories to registries and claims data, thereby enabling the analyses of test results that were otherwise missing. As one example, Roberts et al., 2019 linked data from Genomic Health’s clinical laboratory to SEER data to examine the utilization of a gene expression profiling test for breast cancer (Oncotype DX Breast Recurrence Score) [11].

Two-thirds (67%) of studies focused on cancer, and slightly more than one-third (38%) focused on just two types of commonly used tests (22% of studies focused on gene expression profiling tests for breast cancer and 16% on *BRCA1/2* tests for breast cancer risk) (Table 2). The remaining 33% of studies examined non-cancer conditions, such as metabolic deficiencies, prenatal abnormalities, various neurological conditions, familial hypercholesterolemia, and miscellaneous pharmacogenetic testing. Of the 72 included studies, 22 included the name of the test and laboratory that was utilized to perform the genetic tests (e.g., Genomic Health, GeneSight^®^, Ambry, GeneDx, Myriad, and Foundation Health) (Table 2).

The studies we identified included a variety of research questions. These research questions addressed healthcare utilization (with various perspectives, including geographic, racial, and socioeconomic), the implementation of testing, costs (with various perspectives, including hospital and payer), diagnostic yield, the efficacy of treatment (informed by test results), clinical utility, a description of the referral patterns for test utilization, and appropriate test utilization.

## 4. Discussion

We identified 72 articles that described the use of PM in peer-reviewed RWD studies for a wide variety of tests, nearly 67% of which were cancer-based. Many of these studies were performed using proprietary or subscription-based databases or registries.

We found five categories of databases used in these studies. First, private administrative claims data were frequently used by insurance companies to measure utilization by “claim event.” In this case, the databases were limited to single insurance company claims and were often proprietary or costly to access. Furthermore, in order to accurately characterize the use of PM tests, individual claim events had to be able to be assigned to a specific test, and many PM tests were billed as unlisted or in non-descriptive ways (e.g., CPT81479) as a result. Second, public databases were identified in the form of Medicare claims. Similar to private claims, Medicare claims were also challenging in that they were limited to data collected to the population they served and were subject to billing coding issues. Third, clinical databases, such as medical systems EHR, were commonly used. The prevalence of these datasets could be descriptive of the use of PM at a specific hospital system; however, while such databases are informative, they are only representative of a localized diagnostic and treatment protocol in place at a particular hospital or clinic and may not be depictive of how PM is used across various other types of patient settings or populations. Fourth, laboratory databases, such as those maintained by test manufacturers, were limited to the specific tests offered by the laboratory and were largely proprietary in nature. Additionally, laboratories mainly only offered access to their individual test utilization for clinical studies in conjunction with a research partnership. Last, 55% of RWD used were registries that were created to track a specific test, condition, or population. These types of databases were helpful to study a specific test or condition and were informative, provided that the data included were needed for a particular study. Registries could be limiting if the necessary data were not included and if access was usually only available to particular researchers. Ultimately, each type of database, whether it be private/public claims, clinical, laboratory, or a registry dataset, held advantages and limitations that made each better suited for an understanding of PM utilization across various parts of the United States and the world.

We provide two examples where the use of a database of PM utilization can have clinical, economic, and social implications. We found that nearly 38% of RWD studies on PM were focused on the use of *BRCA1/2* testing or gene expression profiling (GEP). While these tests were important in cancer risk and treatment, they have been in use for many years with established PM utility, and utilization may no longer be tracked. For example, the Surveillance, Epidemiology, and End Results Program (SEER) database is a commonly used database for analyzing breast cancer cases, but the version containing Oncotype Dx data is no longer being updated and only contains data from 2004 to 2016 [12]. On the other hand, tests that are relatively new to PM, such as WES/WGS, were rarely described in the RWD databases (our study found only two such studies), which could have been useful in informing their clinical utility and application. Furthermore, there were many studies of one test type in specific settings that were only informative to that setting or location, and not generalizable across different populations and regions. To demonstrate these last two points, the use of rapid GS has been demonstrated by Dimmock et al., 2021 for diagnosis, change management, and saving money in critically ill infants at five California children’s hospitals, but these data are only found in study documentation and are limited to 184 infants within California’s Medicaid program [13].

Given our findings, there is a great need for a publicly available real-world dataset for PM. This ideal dataset should be cross-cutting in terms of source (e.g., EHR, public/private claims data, and individual registries), the types of conditions (e.g., cancer and non-cancer), and the specific tests used. A dataset, as described, would be beneficial to many stakeholders, such as insurance companies, clinical researchers, and test implementation specialists. The Association of Molecular Pathology Clinical Genomic Data Working Group describes the challenges, opportunities, and solutions surrounding the complexity, lack of standards, fluency, and functionality of genomic data generation, interoperability, and utilization within electronic health records [14]. Beckman and Lew describe the necessity of international data sharing and bioinformatic data consolidation in translating PM into clinically relevant, evidence-based medicine [15]. Additionally, the Professional Society for Health Economics and Outcomes Research (ISPOR) announced that the Real-World Evidence Transparency Initiative has launched the Real-World Evidence Registry [16]. While this registry is open to many types of medical interventions, it is a good launching platform that could inform registries for PM.

### Limitations

Our study has limitations related to our ability to aggregate all appropriate studies in our scoping review. It was infeasible to perform a comprehensive assessment of all studies of PM utilization; specifically, identifying articles that described utilization was difficult due to the lack of “utilization” subject heading or MeSH term in PubMed. Furthermore, we limited our search to articles only found in PubMed, which may limit or show bias toward US publications. Finally, we did not abstract and analyze the sources of funding for each included study. For these reasons, our results are illustrative in nature.

## 5. Conclusions

In conclusion, RWD is essential to understanding the use of PM. Our findings identified studies describing the use of PM in specific settings/tests (e.g., specific EHR, Oncotype Dx) or using specific datasets (e.g., Truvan). As highlighted by others, there are many challenges to using existing databases [4,5,6]. There are no publicly available datasets that can be used to describe the overall use of PM. Future databases and registries are necessary to inform the overall utilization of PM.

## Figures and Tables

**Figure 1 jpm-12-00557-f001:**
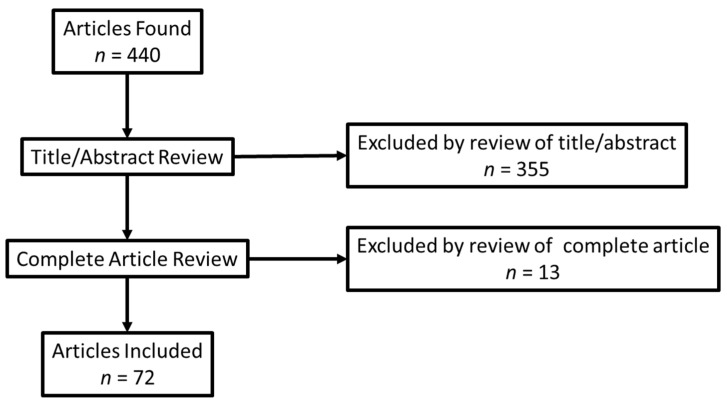
Prisma diagram.

**Table 1 jpm-12-00557-t001:** Percentage of studies by characteristics (*n* = 72).

Real-World Data Type *					Integrated Dataset
Administrative Claims (Commercial)	Administrative Claims (Public)	Clinical Databases	Lab Databases	Registries	Used Integrated Dataset for Analyses
33% 24/72	25% 18/72	29% 21/72	31% 22/72	55% 40/72	38% 27/72

Real-world data type categorizations are derived from Garrison et al. [11]. Registry = prospective, observational data collected longitudinally. * Percentages add up to >100% because studies may have used more than one data source.

**Table 2 jpm-12-00557-t002:** Percentage of test types and examples (*n* = 88) *.

Test	Percentage (*n*) *	Test Examples
*BRCA1* and *BRCA2*	16% (14)	*BRCA1* and *BRCA2*
GEP/Oncotype Dx^®^	22% (19)	Oncotype Dx^®^
Multi-gene panel	10% (9)	Hereditary Cancer Genetic Testing, Myriad MyRisk^®^ Panel
Tumor sequencing	2% (2)	FoundationOne^®^
WES or WGS	2% (2)	Whole Exome or Genome Sequencing
Other	48% (42)	GeneSight^®^ Psychotropic Test, Afirma^®^ CYP450, PGx (undefined)

* *n* is greater than the number of studies (72) as some studies included multiple tests. Abbreviations: GEP, gene expression profiling; WES, whole exome sequencing; WGS, whole genome sequencing.

## Data Availability

Data are available upon request from corresponding author.

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
