# Peer review of "Analyzing Precision Medicine Utilization with Real-World Data: A Scoping Review"

_jpm, 2022, doi:10.3390/jpm12040557_

Round 1

Reviewer 1 Report

The article would be of interest to readers of the journal, and it is very well written. I believe that there are several items that should be addressed before a decision on publication is made, however.

Major comments:

Results are summarized with descriptive statistics in Tables 1 and 2, but nowhere could I find the actual results presented. Ideally there would be one or more tables in the supplemental materials showing exactly how each of the 72 included articles was coded. Categories in both Table 1 and Table 2 are not mutually exclusive, and as presented it is not possible for readers to tell where overlap exists. Presenting the full data as coded would also contribute to the potential reproducibility of the work.  

While a PRISMA flowchart is presented, it does not appear that the Preferred Reporting Items for Systematic reviews and Meta-Analyses extension for Scoping Reviews (PRISMA-ScR) Checklist was used. If it was used, please say so and present the completed checklist as a part of the supplemental materials. If it was not used, then this could be listed in the limitations section.

PubMed was the only database searched, and this is not mentioned in the limitations section. In addition to the potential to miss relevant publications by not searching additional databases, there may be a bias towards US publications associated with the use of PubMed only.

Minor issues:

Line 21 – I think “Slightly more than two-thirds (38%)” should be “Slightly more than one-third (38%)”

Line 29 - The Human Genome Project concluded in 2003, however, this does not mean that the first human genome was sequenced in 2003.

Line 31 - Cited reference [1] (Gameiro et al., 2018) does not adequately support "exponential reduction in cost of genomic sequencing over the last decade." Gameiro et al. does say that "Ten years ago, sequencing the genome cost US$ 1 billion and took 13 years; today it costs US$ 1,500 and takes only a few hours," however, no reference is cited by Gameriro to support this. National Human Genome Research Institute (genome.gov/about-genomics/fact-sheets/Sequencing-Human-Genome-cost), reports that cost of sequencing a human genome dropped from about $14 million to roughly $1500 over the period from 2006-2015. While that is an impressive reduction, just saying cost dropped exponentially is uninformative regarding how much sequencing actually costs today. The rate of reduction in cost from 2006-2015 has slowed over the period from 2015-2021. For a paper submitted in 2022, the past decade would mean 2012-2021.

Line 39 – contraction (it’s) should be written (it is)

Lines 78-80 - The use of e.g., in the definitions used for inclusion/exclusion criteria is problematic, especially in the case of the RWD definition. The reference cited as source for this definition mentions patient-generated data including in home-use settings, and data gathered from other sources that can inform on health status, such as mobile devices. The definition for RWD provided in the submitted article does not make clear whether patient-generated data and mobile devices would or would not have been eligible for inclusion.

Author Response

Reviewer comment: The article would be of interest to readers of the journal, and it is very well written. I believe that there are several items that should be addressed before a decision on publication is made, however.

Major comments:

Results are summarized with descriptive statistics in Tables 1 and 2, but nowhere could I find the actual results presented. Ideally there would be one or more tables in the supplemental materials showing exactly how each of the 72 included articles was coded. Categories in both Table 1 and Table 2 are not mutually exclusive, and as presented it is not possible for readers to tell where overlap exists. Presenting the full data as coded would also contribute to the potential reproducibility of the work.  

Author response: We have provided a table in the supplemental materials with the coding for each of the 72 articles.

Reviewer comment: While a PRISMA flowchart is presented, it does not appear that the Preferred Reporting Items for Systematic reviews and Meta-Analyses extension for Scoping Reviews (PRISMA-ScR) Checklist was used. If it was used, please say so and present the completed checklist as a part of the supplemental materials. If it was not used, then this could be listed in the limitations section.

Author response: We have included the PRISMA-ScR checklist in the supplemental materials.

Reviewer comment: PubMed was the only database searched, and this is not mentioned in the limitations section. In addition to the potential to miss relevant publications by not searching additional databases, there may be a bias towards US publications associated with the use of PubMed only.

Author response: We have included a statement to this effect in the limitations section.

Minor issues:

Reviewer comment: Line 21 – I think “Slightly more than two-thirds (38%)” should be “Slightly more than one-third (38%)”

Author response: This was an error and has been corrected.

Reviewer comment: Line 29 - The Human Genome Project concluded in 2003, however, this does not mean that the first human genome was sequenced in 2003.

Author response: Agree, the degree of completion varies and is subjective. We have removed the date.

Reviewer comment: Line 31 - Cited reference [1] (Gameiro et al., 2018) does not adequately support "exponential reduction in cost of genomic sequencing over the last decade." Gameiro et al. does say that "Ten years ago, sequencing the genome cost US$ 1 billion and took 13 years; today it costs US$ 1,500 and takes only a few hours," however, no reference is cited by Gameriro to support this. National Human Genome Research Institute (genome.gov/about-genomics/fact-sheets/Sequencing-Human-Genome-cost), reports that cost of sequencing a human genome dropped from about $14 million to roughly $1500 over the period from 2006-2015. While that is an impressive reduction, just saying cost dropped exponentially is uninformative regarding how much sequencing actually costs today. The rate of reduction in cost from 2006-2015 has slowed over the period from 2015-2021. For a paper submitted in 2022, the past decade would mean 2012-2021.

Author response: Agree, “exponential” is an uninformative term and we have revised the language to be specific per the NHGRI DNA Costs website and referenced the website as the source.

Reviewer comment: Line 39 – contraction (it’s) should be written (it is)

Author response: Corrected

Reviewer comment: Lines 78-80 - The use of e.g., in the definitions used for inclusion/exclusion criteria is problematic, especially in the case of the RWD definition. The reference cited as source for this definition mentions patient-generated data including in home-use settings, and data gathered from other sources that can inform on health status, such as mobile devices. The definition for RWD provided in the submitted article does not make clear whether patient-generated data and mobile devices would or would not have been eligible for inclusion.

Author response: We have removed the reference and clarified what is included in our definition of RWD.

Reviewer 2 Report

The manuscript aims to clarify the characteristics of real-world data (RWD) sources related to precision medicine using a scoping review. This attempt could certainly play a key role in understanding specific practical initiatives in precision medicine and in considering the future nature of precision medicine and should be encouraged and developed. However, the current article has at least three major drawbacks.

First, while the manuscript clearly describes the process and result of the intended scoping review, it does not seem well organized. A better structure of the overall manuscript is recommended: e.g., in the “Introduction,” the connections and flows between sentences in paragraphs and between paragraphs should be reconsidered, and in the “Limitations” and “Conclusions,” the descriptions should be more concretized.

Second, in the “Introduction,” “Discussion,” and “Conclusion,” the authors should emphasize the strength (i.e., novelty and uniqueness) of this research using the analysis of RWD sources. This would include specific knowledge gaps from previous studies and perspectives, emerging challenges, and possible suggestions for the proactive practice of using RWD sources as well as the clinical, economic, and social implications based on the current precision medicine practices.

Lastly, data registration incentives and data quality assurance of RWD sources as well as the potential gap between RWD sources in the selected peer-reviewed articles and the real practices in precision medicine should be mentioned in the manuscript. Regarding the latter, one should therefore consider if there could be some bias/tendency in precision medicine practices that either promotes or precludes their registration and publication.

Specific comments:

Abstract

p.1, l.21

The phrase “two-thirds” should be changed to “one-third.”

1. Introduction

p.1, l.39–40

It is unclear what the current challenge is when using RWD.

2. Material and Methods

p.2, l.72

Is there any particular reason why the scope of this research was initiated from January 2021? If there is a justification, please provide this.

2.2. Data Abstraction

p.2, l.87

Could you briefly describe what kind of discrepancies have arisen in the analysis?

3. Result

p.3, l.119–122

Could you confirm the connection between the percentages (e.g., 67%, 44%, and 25%) described in the text and the percentages shown in Table 2? While these differences seem to have arisen because the number of tests (N = 88) and the number of included studies (N = 72) is different, it is difficult to understand the meaning of the percentages in the text relative to those in Table 2.

Author Response

Reviewer comment: The manuscript aims to clarify the characteristics of real-world data (RWD) sources related to precision medicine using a scoping review. This attempt could certainly play a key role in understanding specific practical initiatives in precision medicine and in considering the future nature of precision medicine and should be encouraged and developed. However, the current article has at least three major drawbacks.

First, while the manuscript clearly describes the process and result of the intended scoping review, it does not seem well organized. A better structure of the overall manuscript is recommended: e.g., in the “Introduction,” the connections and flows between sentences in paragraphs and between paragraphs should be reconsidered, and in the “Limitations” and “Conclusions,” the descriptions should be more concretized.

Author response: We have revised the manuscript to better lead the reader between sentences and paragraphs. We have also expanded the limitations and conclusions with more concrete descriptions or examples.

Reviewer comment: Second, in the “Introduction,” “Discussion,” and “Conclusion,” the authors should emphasize the strength (i.e., novelty and uniqueness) of this research using the analysis of RWD sources. This would include specific knowledge gaps from previous studies and perspectives, emerging challenges, and possible suggestions for the proactive practice of using RWD sources as well as the clinical, economic, and social implications based on the current precision medicine practices.

Author response: We have expanded the introduction and conclusion to include examples of the challenges to using existing RWD. We include a paragraph describing the need for a publicly available RWD and what this ideal dataset may include.We have expanded the discussion to include examples that demonstrate the lack of databases that could use RWD to demonstrate clinical, economic, and social implications.

Reviewer comment: Lastly, data registration incentives and data quality assurance of RWD sources as well as the potential gap between RWD sources in the selected peer-reviewed articles and the real practices in precision medicine should be mentioned in the manuscript. Regarding the latter, one should therefore consider if there could be some bias/tendency in precision medicine practices that either promotes or precludes their registration and publication.

Author response: We recognize the concern from the reviewer. We have included the PRISMA-ScR checklist in the supplemental materials noting that we did not appraise the individual sources of evidence. An appraisal of this nature was beyond the scope of this paper which was to identify databases containing PM. We have noted in the manuscript the challenges of using RWD.

Specific comments:

Reviewer comment: Abstract: p.1, l.21. The phrase “two-thirds” should be changed to “one-third.”

Author response: This was an error and has been corrected.

Reviewer comment: 1. Introduction: p.1, l.39–40. It is unclear what the current challenge is when using RWD.

Author response: We have defined the challenges to using RWD.

Reviewer comment: 2. Material and Methods: p.2, l.72. Is there any particular reason why the scope of this research was initiated from January 2021? If there is a justification, please provide this.

Author response: The scope was initiated from January 2015 and we have included a justification in the materials and methods. “We chose to limit our search to 5 years plus the current year as this would identify the majority of articles on the topic (only 9 articles would have been identified in the search prior to 2015, none of which met inclusion criteria).”

Reviewer comment: 2.2. Data Abstraction: p.2, l.87. Could you briefly describe what kind of discrepancies have arisen in the analysis?

Author response: We have included some examples.

Reviewer comment: 3. Result: p.3, l.119–122. Could you confirm the connection between the percentages (e.g., 67%, 44%, and 25%) described in the text and the percentages shown in Table 2? While these differences seem to have arisen because the number of tests (N = 88) and the number of included studies (N = 72) is different, it is difficult to understand the meaning of the percentages in the text relative to those in Table 2.

Author response: We have clarified the % in the test as relative to the % in the table. There was an error in how the percentages were categorized. We have recalculated all data and percentages in the manuscript and corrected the error throughout the manuscript.

Round 2

Reviewer 1 Report

In this revision, the authors have addressed all of the concerns I had in my previous review. I believe that the manuscript is now acceptable for publication.

Author Response

In this revision, the authors have addressed all of the concerns I had in my previous review. I believe that the manuscript is now acceptable for publication.

  • Thank you for your comments and input to make the manuscript stronger.

Reviewer 2 Report

I thank you for your rapid response and kind consideration. I have reviewed the authors’ responses and their revised manuscript. I believe that the authors’ proactiveness has contributed to increasing the value of this manuscript. While I truly appreciate the additional effort, to approve the manuscript, I request the authors to provide final confirmation of the supplementary materials (S3) (in particular, to maintain the consistent style (e.g., "Research Question," "Clinical Details," "Admin Claims," and "Registry Details")) and resubmit it.

Author Response

I thank you for your rapid response and kind consideration. I have reviewed the authors’ responses and their revised manuscript. I believe that the authors’ proactiveness has contributed to increasing the value of this manuscript. While I truly appreciate the additional effort, to approve the manuscript, I request the authors to provide final confirmation of the supplementary materials (S3) (in particular, to maintain the consistent style (e.g., "Research Question," "Clinical Details," "Admin Claims," and "Registry Details")) and resubmit it.

  • Thank you for your comments. We have included an Excel file with the abstracted data and coding. Additionally, in the methods section we have confirmed how the studies were organized and indicated the header for the types of coding ensuring the names are consistent in the manuscript and the spreadsheet.
  • Please note the above required use to reorder the items in the supplemental materials.